# *Lactobacillus reuteri* DSM 17938 as a Novel Topical Cosmetic Ingredient: A Proof of Concept Clinical Study in Adults with Atopic Dermatitis

**DOI:** 10.3390/microorganisms8071026

**Published:** 2020-07-11

**Authors:** Éile Butler, Christoffer Lundqvist, Jakob Axelsson

**Affiliations:** 1BioGaia AB, Mobilvägen 10, 223 62 Lund, Sweden; eb@biogaia.se (É.B.); cl@biogaia.se (C.L.); 2Faculty of health and society, Department of Health Biomedical, Malmö University, Jan Waldenströms Gata 25, 214 28 Malmö, Sweden

**Keywords:** *Lactobacillus reuteri* DSM 17938, cosmetic, probiotics, atopic dermatitis, adults, in-use test, topical application

## Abstract

Atopic Dermatitis (AD) is a chronically relapsing skin condition characterized by dry, itchy, and inflamed skin where sufferers can frequently be subject to infections. Probiotics are known to be potent immune-modulators, and live *Lactobacillus reuteri* DSM 17938 has shown to be anti-inflammatory but also to possess antimicrobial and barrier function properties. This study aimed to investigate and compare two investigational ointment products (topical probiotic and control) for cutaneous acceptability, safety, and efficacy under normal conditions of use, in adult subjects with atopic dermatitis. The products were applied twice daily for 8 weeks, and cutaneous acceptability, SCORAD index, local SCORAD, and adverse events were evaluated after 4 and 8 weeks of treatment. At the end of the observations, it was demonstrated that both the probiotic-containing and probiotic-free ointments were both cutaneously acceptable and safe. It importantly showed a statistically and clinically significant improvement of the SCORAD index and local SCORAD in adult subjects with AD after 4 and 8 weeks of continuous use. In conclusion, we show evidence that the probiotic product, containing live *L. reuteri* DSM 17938 as an extra ingredient, is safe and promising as a novel topical cosmetic ointment and with further testing could be a standard topical product for the management of atopic dermatitis or other disorders associated with the skin.

## 1. Introduction

Atopic Dermatitis (AD) is a chronically relapsing skin condition characterized by dry, itchy, and inflamed skin where sufferers can frequently be subject to infections [1]. According to the International Study of Asthma and Allergies in Childhood (ISAAC), the prevalence of AD in children varies significantly from 0.3% to 20.5% among 56 countries [2] with the condition usually clearing up as the child ages. However, the condition can continue into adulthood, causing a substantial economic and health burden for patients and health services. For example, in the United States, there is an estimated 3.5 to 7.2% prevalence of adults living with AD [3,4]. The mechanism behind AD can be seen as multifactorial, where allergens from food and the environment, as well as genetics, have a part to play in AD development and manifestation. Previous focus has been on underlying immunological mechanisms, but more research is now emerging on skin barrier defects and impairments characteristic of AD [5,6]. Recently, the skin microbiome has also been implicated in AD, a dysbiosis in the skin microbiome, and increased *Staphylococcus aureus* colonization has been shown to correlate with increased lesion infection and is an exacerbating factor of the condition [7,8,9].

In all cases, topical ointments are essential in the long term management of the condition through protecting and restoring the skin barrier, reducing water loss, and decreasing inflammation and symptoms like itching [10]. Moderate to severe cases of AD are usually treated with anti-inflammatory agents such as topical corticosteroids [11], as well as continuous topical ointment use [12,13]. These treatments have long-term side effects, and many sufferers are searching for safe and efficient products that can lengthen the time between flareups, protect the skin barrier, as well as reducing the need for topical corticosteroid use [14,15]. Product development, including the use of plant-based oils and probiotics, are attractive alternatives with few risks of side effects [16,17]. Probiotics, defined as “live microorganisms that, when administered in adequate amounts, confer a health benefit on the host” [18], have been used for decades as dietary supplements in the area of gut health and now many researchers are investigating how orally administered probiotics can influence the skin with promising results in many areas, including AD [19,20].

In recent years probiotics and their derivatives (lysates, postbiotics) have also begun to be investigated for topical use in terms of safety and efficacy on the skin in subjects with sensitive or reactive skin [21,22], AD [23,24] or acne-prone skin [25]. Multiple bacterial strains have been investigated and have been shown to have anti-inflammatory, antimicrobial, and barrier function properties, which have benefits in different skin conditions [26,27,28].

*Lactobacillus reuteri* is one of the most widely studied probiotic strains in many areas of human health [29,30]. *L. reuteri* DSM 17938 is a commercially available probiotic of human origin, that has been researched extensively in the area of gastrointestinal health, inflammation, and pathogen inhibition [31,32]. This probiotic strain is well researched from a safety perspective and has been used in numerous human clinical trials, a detailed safety profile on blood cell counts, and inflammation markers [33]. Furthermore, it has been cured of two plasmids carrying antibiotics resistance genes from its parent strain *L. reuteri* ATCC 55730 [34]. It is capable of producing antimicrobial substances, such as reuterin, which inhibits both Gram-positive and Gram-negative bacteria [35,36], as well as suppressing relevant pathogens [26,37]. *L. reuteri* DSM 17938 and its mother strain have anti-inflammatory properties, such as suppressing TNF-a response in innate immune cells [38], as well as immunomodulatory properties manifested by increased proliferation of regulatory T lymphocytes [39,40]. It is shown to have barrier enhancing capabilities by competitive exclusion of *S. aureus* in the skin [37,41], and by increasing tight junction proteins in the gut [42]. The combined effects of all these properties, and being one of the most widely studied probiotics with proven benefits for patients suffering from IgE-associated eczema when administered perorally [20], make it a probiotic strain very attractive for further investigations. Many products available on the market contain inactivated forms of probiotic strains. However, this strain displayed pronounced anti-inflammatory and barrier function properties in an ex vivo skin model, and in vitro antimicrobial activity against opportunistic skin pathogens compared to its lysed form [37].

We, therefore, hypothesize that live *L. reuteri* DSM 17938 added as an extra ingredient could have a beneficial effect on the skin when added to a novel barrier protecting ointment in adults with AD or dry, inflamed skin. This study aimed to investigate and compare a topical ointment product with and without live probiotic bacteria for cutaneous acceptability, safety, and efficacy under normal conditions of use, in adult subjects with atopic dermatitis.

## 2. Materials and Methods

### 2.1. Ethics Statement

The study was a randomized, double-blind controlled study under dermatological control. The protocol complied with the ethical guidelines of the 1975 Declaration of Helsinki and was approved by the Pharma Ethics board in South Africa (Reference 180,821,082) before the study started. The study was also registered on www.clinicaltrials.gov (NCT03632174). Written informed consent was obtained from each subject before enrollment.

### 2.2. Investigational Products

Two products were manufactured in Sweden and shipped in refrigerated conditions to South Africa for the study. The probiotic product was an ointment containing shea butter, canola oil, hydrogenated canola oil, and live *L. reuteri* DSM 17938 contained in a 25 g aluminum tube (BioGaia AB, Lund, Sweden). The live but freeze-dried probiotic strain was added to the formulation to reach a concentration of a minimum 1 × 10^8^ CFU/gram throughout the study period. The product was evaluated for stability for the study duration at both 25 °C and 5 °C to monitor the stability of the bacteria using standardized laboratory protocols.

The control product was identical but without the live probiotic. The odor, color, and consistency of the two products were the same. Subjects were randomized to apply either probiotic product or control product and were instructed to apply the ointment on the affected areas of the whole body twice per day (morning and evening), for 8 weeks.

Subjects were given enough tubes for the study duration, weighed for each subject at the beginning and end of the study. The tubes were stored at 5 °C until supplied to the subjects. Products were randomized using a computer-generated randomization list (1:1).

### 2.3. Patient Recruitment and Sampling

The study included 36 subjects and was performed between January and August 2019 in Cape Town, South Africa. The subjects were aged between 18–70 years old with AD, according to the definition of the U.K. Working Party’s Diagnostic Criteria for Atopic Dermatitis, SCORing Atopic Dermatitis (SCORAD) index of >25 at the baseline (day 0) [43]. Other specific inclusion criteria included subjects presenting with a current lesion of AD on a defined area. Each subject had a two week run-in period where the subject could not modify their cosmetic/washing habits, use their standard topical treatment or apply cosmetics on the concerned areas. Subjects were restricted from using their other standard treatments for AD, using antihistamines and using certain medical treatments. They were instructed not to change their hygiene practices, to apply any other topical products, as well as to avoid sun exposure outside their usual habits for the duration of the study. Each subject participated in the study for 8 weeks and came for 3 study visits; baseline (day 0), day 28 (visit 2), and day 56 (visit 3).

Exclusion criteria stated that the subjects should not be involved in other biomedical research studies at the same time as the study, subjects who had not respected the run-in period, subjects who changed their cosmetic habits within the two weeks before the study start, or subjects who applied their cosmetics or other pharmaceutical products on the day of inclusion into the study. In addition, exclusion criteria included immunological, cardiovascular, pulmonary, digestive, neurologic, psychiatric, genital, urinary, hematological, or endocrine disease or under immunosuppressive or long term-treatment of aspirin, anti-inflammatory agents, antibiotics, antihistamines, corticoids, beta-blocker, and desensitization drugs.

The primary endpoint was to evaluate the cutaneous acceptability of the products, which was performed by the dermatologist investigator through a physical examination (baseline and visit 2), and by the subjects through completion of a questionnaire (baseline, visit 2). The secondary endpoint was the improvement of skin appearance evaluated by the dermatologist through SCORAD index and local SCORAD (baseline, visit 2, visit 3), and cosmetic qualities and efficacy through questionnaires filled in by the subject in the presence of the dermatologist on visit 2. Both the local SCORAD and SCORAD index were quantified to include both a lesion of defined areas, as well as more generalized events from the AD. Adverse events were also monitored throughout the study.

### 2.4. Appraisal of the Cutaneous Acceptability by the Dermatologist Investigator and Cosmetic Acceptability

The cutaneous acceptability of cosmetic products was assessed as previously described in other studies [44]. The clinical examination was carried out at baseline by the dermatologist investigator before the start of the study and after 4 weeks (visit 2) of continuous application. The cutaneous acceptability was assessed by firstly observing physical signs linked directly to the investigational product through nature, location, and sensitivity. Secondly, it was assessed through a questionnaire filled in by each subject on the functional signs (prickling, tightening, heating sensation) linked to the product. The questionnaire was filled in at home daily by the subject detailing application frequency and the number of applications, as well as experienced reactions (if any). Cosmetic acceptability was evaluated using standardized evaluation procedures [45]. Appraisal of the cosmetic acceptability and efficacy of the products by the participant was assessed through questionnaires adapted to the investigational products in the presence of the study monitor, and it was filled in on the final visit by the subject (scale: very good, good, rather good, mediocre, bad). The questionnaire was completed by an appraisal of the products in the presence of the dermatologist.

### 2.5. Clinical Evaluation by the Dermatologist Investigator

SCORAD was evaluated using the standardized form from the European task force on Atopic Dermatitis [46]. Evaluation of each parameter of SCORAD index, which includes the sum of the total involved area, the intensity of the lesions (xerosis, erythema, excoriation), and subjective symptoms (itching, sleep loss), for each subject, was performed before the start of use, after 4 weeks (visit 2) of application and after 8 weeks (visit 3) of the application by the dermatologist. Local SCORAD (score of intensity of the physical signs) was evaluated by the dermatologist investigator, at each visit (baseline, visit 2 and visit 3), on the recurrent lesion, and on the control area that was selected at baseline (day 0).

### 2.6. Determination of Number of Subject’s Sample Size

This study is a proof of concept study. At least 20 subjects in each group are frequently used for in-use tests, and it is considered as a sufficient number to reach a conclusion.

### 2.7. Statistical Analysis

Data were analyzed using the Intention to Treat approach (ITT). Quantitative variables were summarized as means and standard deviations or median and interquartile range, and categorical variables were summarized with proportions. No adjustment for multiplicity was made, and nominal *p*-values are presented. A *p*-value below 0.05 was considered significant.

The Wilcoxon test was used to evaluate the time effect of the products between baseline, visit 2, and visit 3. The Mann–Whitney test was used to evaluate if there were differences between the two products from baseline, visit 2, and visit 3 (end of study). The statistical analysis was performed using R software (R Foundation for Statistical Computing, Vienna, Austria) (version used 3.2.1). CONSORT recommendations were followed for the reporting of clinical trials (Figure A1).

### 2.8. Adverse Events

Adverse events were reported as outlined in the ICH guidelines for Good Clinical Practice E6 (R2).

## 3. Results

The anticipated number of study subjects was 40 (20 subjects per group); however, the final number recruited was 36 due to protocol violations. Additionally, two subjects missed visit 2 or visit 3 (however, the visit they did attend was included in the calculations). The final evaluation was, therefore, performed on 34 subjects, 17 subjects per group (Figure A1).

### 3.1. Subject Characteristics

The characteristics of each group of subjects can be seen in Table 1. The number of subjects in each group was equal, and the mean age of each group was close to similar. Ethnicity was also similar between each group; 65% of the subjects were black in the control group, while 58% were black in the probiotic group (Table 1). At baseline, there was a slight difference between the SCORAD index and local SCORAD; however, there was no significant difference between the scores at day 0 (*p* = 0.512)). The average volume of product used after 8 weeks was 62 g, with an overall range between 33.1 g–128.1 g from the 30 individuals who returned the products at the end of the study. Stability of the *L. reuteri* DSM 17938 was monitored in the product at the sponsor’s laboratory using validated methods for the duration of the study period and was stable (data not shown).

### 3.2. Cutaneous Acceptability of the Investigational Products

For the probiotic-containing product, the results obtained from the dermatologist investigator’s evaluation and questionnaires revealed very good acceptability of the investigational product in 10 out of 17 of the subjects. For three out of the other seven subjects, it was observed by the dermatologist, and there was moderate erosion for one subject, slight erythema, erosion and edema for the second subject, and slight edema and dryness for the third subject (Table A1). Seven of the subjects indicated having felt discomfort and presented with redness or dryness of slight to moderate intensity. The intensity, duration, and frequency of the appearance of these reactions are frequently encountered with this type of investigational product studied under normal conditions of use.

For the control product, the results obtained revealed very good acceptability of the investigational product in 12 out of 17 of the subjects. The other five subjects indicated having felt discomfort and having presented with redness and dryness or small pimples of slight to moderate intensity. The intensity, duration, and frequency of the appearance of these reactions are frequently encountered with this type of investigational product during this type of study under normal conditions of use.

The subjects also concluded that the products were acceptable (Table A1). The overall analysis of the reactions observed for these two investigational products concluded that the acceptability level was equivalent for both products, and they were acceptable for use in adults with AD.

### 3.3. The Effect of the Products on SCORAD from Baseline to End of Study (Time Effect)

The total number of subjects analyzed for the efficacy of the two products was *n* = 33. SCORAD index and local SCORAD were calculated at baseline, day 28 (visit 2), and day 56 (visit 3) on 17 subjects in the probiotic group and 16 subjects in the control group (one subject for day 28, and another subject for day 56) (Figure 1 and Figure 2 and Table A2 and Table A3).

There was a mean reduction of the SCORAD index of −11.54 (28%) at visit 2, and a mean reduction of −19.06 (46%) at visit 3 in the group using the probiotic product, showing a significant group effect over time from baseline (*p* ≤ 0.001) (Figure 1 and Table A2). There was a mean reduction of the SCORAD index of −11.71 (32%) at visit 2 and a mean reduction of −15.04 (39%) at visit 3 (Day 56) in the group using control product showing a significant time effect from baseline (*p* ≤ 0.001) (Figure 1 and Table A2). The probiotic product showed a tendency to reduce SCORAD index compared to the control. However, this result was not significant (Figure 1).

### 3.4. The Effect of the Products on Local SCORAD from Baseline to End of Study (Time Effect)

There was a mean reduction of −2.24 (25%) of the local SCORAD at visit 2 (Day 28), and a mean reduction of −4.06 (45%) at visit 3 (Day 56) (*p* = <0.001) in the group using the probiotic, showing a significant time effect (Figure 2 and Table A3). There was a mean reduction of −2.81 (34%) of the local SCORAD index at day 28 and a decrease of −3.75 (43%) at Day 56 in the group using the control product, showing a significant time effect from baseline (*p* = 0.001) (Figure 2 and Table A3). When comparing the control to the probiotic, there was no significant difference in the overall effect between the groups (Figure 2 and Table A3). The qualitative effect of the probiotic product can be seen in Figure 3, where before (A), during (B), and after (C) photos of the lesional area of a subject are displayed. The local SCORAD decreased from a starting value 8 to 2 in subject 6, a value of 14 to 5 in subjects 24 and 6 to 0 in subject 36 after eight weeks of continuous use (Figure 3).

Although no significant difference was observed between the two products for the total SCORAD index, some trends were observed. Within the SCORAD-intensity of lesions score (Sum of Erythema, Edema, Oozing, Excoriation, Lichenification, and Dryness) a stronger reduction in the dryness of the probiotic product compared to the control after 8 weeks of continuous use was observed but was not significant (Figure 4a) (*p* = 0.102). A positive trend was also seen for dryness for local SCORAD, where the probiotic product showed a mean variation of 0.46 (24%) in reducing dryness (*p* = 0.062) compared to the control product (Figure 4b).

### 3.5. Adverse Events

There were no adverse events observed in the duration of this study.

## 4. Discussion

This clinical trial shows that an ointment containing probiotics is well tolerated and effective in reducing AD symptoms. Two products were investigated, one active product containing a live probiotic, *L. reuteri* DSM 17938, which has previously been shown to have antimicrobial and anti-inflammatory capabilities [37] and a control product without the probiotics. Both products contain ingredients commonly used in topical AD ointments possessing anti-inflammatory, barrier hydrating/enhancing properties, as well as decreasing *S. aureus* colonization [12,47,48,49]. The products tested in this study contained shea butter, which has been shown to have anti-inflammatory and antioxidant properties and is widely used in cosmetic products for treating dry skin and dermatitis [17]. AD affects the quality of life, both physically and emotionally, in subjects who suffer from the disorder, and topical ointments are extremely important in its management [14].

These two products in this study used on subjects with AD were both concluded to be cutaneously acceptable by the dermatologist investigator Table A1 (Appendix A). Although some reactions were noted, it can be concluded that these are quite normal when using such products on subjects with such a condition, and can occur in some individuals with sensitive skin when applying new products [50]. We can observe from these results that *L. reuteri* DSM 17938 as an additive ingredient to the topical ointment does not affect the cosmetic or cutaneous acceptability of the product. As well as this, AD is a multifactorial condition that usually needs multiple management steps to manage the condition, including specific bathing instructions, clothing, and avoiding allergens. In the case of topical products, the product must be suitable for use on the skin; therefore it is important to understand that the control product, in this case, is an effective product in itself in the management of AD, Table A1 (Appendix A), as it contains other barrier protecting ingredients, like shea butter [17,51].

We observed in this study that both products had a statistical and clinically significant improvement of the SCORAD index after 4 weeks and 8 weeks of use based on evaluations by the dermatologist investigator (Figure 1). Although the probiotic product showed a slightly greater percentage-wise improvement in symptoms of the SCORAD after 8 weeks compared to the control product (−46% and −45% for the probiotic compared to −39% and −43% for the control), this difference was not statistically significant. This was similar to a significant improvement in local-SCORAD after 8 weeks (Figure 2) that was not significantly different between the products. This could be due to the low statistical power of the study (*n* = 34) or the length of treatment time.

In this study, we saw that there were some promising trends toward *L. reuteri* DSM 17938 containing ointment to decrease dryness compared to the product without it (Figure 4a,b). Hydration of the skin helps to improve dryness, reduce pruritus, and restore disturbed barrier function [51]. It has been shown that ointment use can strengthen the skin barrier in atopic dermatitis by restoring hydration to the stratum corneum and reducing trans-epidermal water loss [14,15]. In pre-clinical experiments, it was observed that *L. reuteri* DSM 17938 had an effect on barrier function genes such as aquaporin 3, associated with skin hydration [37], and is overexpressed in AD skin [52]. Further studies should be performed to investigate if the probiotic product used in this could reduce dryness and moisture loss in AD subjects through the upregulation of aquaporin 3 or other barrier function genes.

Both products investigated in this study could significantly reduce itching (−58% probiotic and −34% for control) and sleep loss (−78% for probiotic and −76% control) over time from baseline to end of the study. However, the difference was not significant between the two products (data not shown). This was most likely due to a significant difference between the starting SCORAD values for itching between both the probiotic and control product (*p* = 0.017, data not shown). Subjective symptoms such as itching and sleep loss are extremely significant and debilitating for subjects with AD [53]; therefore, using ointments that can manage these symptoms is essential.

Due to these results and trends, it is reasonable to assume that with larger sample size and longer product administration, that the probiotic product could show a significant difference. *L. reuteri* species have previously been shown to affect IgE associated eczema [20], immunomodulation, and pathogen inhibition [38]. They have recently been shown to interact with pain receptors such as TRPV1 receptor in the gut [54]. Interestingly the latter is a prominent receptor for itch and is expressed on epidermal keratinocytes and other cells found in the skin and has been discussed as an important molecule in itching signaling in AD [55].

Further studies should be performed to investigate if topical application of *L. reuteri* DSM 17938 could reduce itching through different receptors and inflammatory pathways in the skin, including TRPV1.

In this present study, no skin microbiota samples were taken. It is now known that the skin microbiota is a significant part of the skin and has an important role in its protection through its interaction with different skin cells, barrier function genes, the production of chemical modulators, and defending against invading pathogenic or opportunistic bacteria [56,57]. Other studies have shown the effect of bacterial ingredients on AD. For example a study by La Roche Posay in adult subjects with AD that investigated the use of a topical ointment containing a non-pathogenic bacteria (*Vitreoscilla filiformis*) reported that in comparison to another ointment, it was able to modify the skin microbiome and reduce the number of flareups and the severity of the AD [47]. When the microbiome becomes imbalanced, there can be a breakdown of the barrier, an increase in inflammation and infection leading to disease such as AD, psoriasis, wounds, and acne [56]. In particular, subjects with AD are highly colonized by *S. aureus* [58] and show a loss in bacterial diversity on the skin [59], and another study observed that *S. aureus* colonization is associated with more severe AD in children [60].

As mentioned, *L. reuteri* DSM 17938 is a widely researched and characterized probiotic strain that has been shown to have significant anti-inflammatory and pathogen inhibitory effects. However, this is the first time, to our knowledge, that it has been combined in a topical ointment for use in subjects with AD. As well as this, due to some promising results using oral probiotics, a complementary approach could also be of interest.

We have previously shown that *L. reuteri* DSM 17938 can inhibit opportunistic and pathogenic bacteria associated with the skin, such as *S. aureus* [37]. This inhibition can be both a direct effect of *L. reuteri* DSM 17938 secreting antimicrobial substances, such as reuterin [35,36], and competitive exclusion. Research has also shown that *L. reuteri* ATCC 55730, the mother strain of *L. reuteri* DSM 17938, can decrease adherence by competing against *S. aureus* in an in-vitro model, and thereby protecting keratinocytes from *S. aureus* through competitive exclusion [26]. Other Lactobacilli, such as *L. rhamnosus* GG, have been shown to inhibit adhesion of pathogens by inhibition if pili interactions [61]. However, no studies have investigated how *L. reuteri* affects and interacts with the skin microbiota but it will be important in understanding its effects on skin disorders and its effect on healthy skin. *L. reuteri* DSM 17938 has been shown to inhibit various pathogens in vitro and thereby is capable of influencing the microbiota, as well as tolerate aerobic conditions such as on the skin, making it an interesting candidate for further investigations into microbiome modulation.

Although this study observed interesting and novel results, there were some limitations. Firstly, the recruitment period was quite long from January to June, and this could affect the treatment results as atopic dermatitis symptoms can vary between seasons in their severity. One reason for the long recruitment period was likely the criteria of a two-week run-in period when the subjects were not allowed to use their normal standard topical treatments making many subjects reluctant to participate due to the symptoms associated with AD.

Secondly, the sex of the subjects in this study was primarily female; therefore, including both males and females equally in a follow-up study would be important. Thirdly, the broad inclusion criteria (no upper limit on baseline SCORAD index for inclusion) resulted in increased variation of the results compared to a more homogenous study population.

Finally, other analyses, such as the skin microbiome, were not investigated. This could have given a clue to the health of the skin and some interesting feedback on how the topical products interacted with the microbiota.

## 5. Conclusions

This study is a proof of concept study, demonstrating that a probiotic containing ointment is both cutaneously acceptable and safe for use on the skin, and importantly showed a statistically and clinically significant improvement of the SCORAD index and local SCORAD in adult subjects with AD after 4 and 8 weeks of continuous use.

In conclusion, the results suggest that the probiotic product, containing live *L. reuteri* DSM 17938 as an extra ingredient, shows promise as a novel topical cosmetic ointment and with further testing could be a standard topical product for the management of atopic dermatitis or other conditions associated with the skin.

## Figures and Tables

**Figure 1 microorganisms-08-01026-f001:**
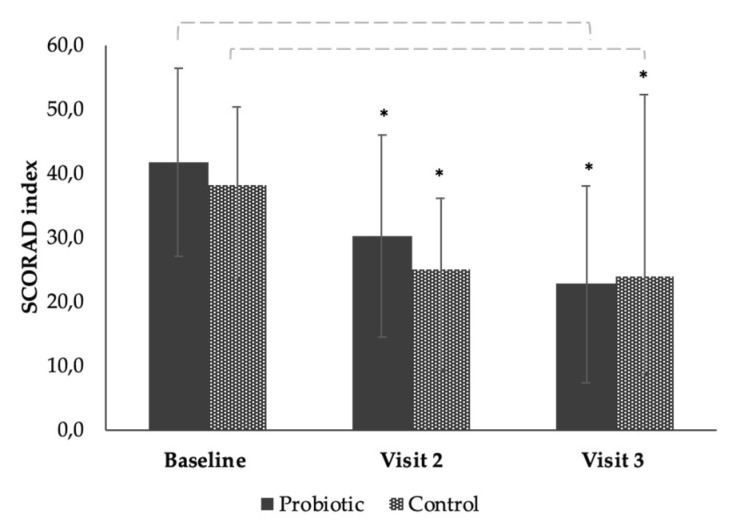
Bar chart showing the mean reduction in SCORAD index of the subjects after continuous use of the probiotic or control product at baseline, visit 2 (day 28), and end of study visit 3 (day 56). Each error bar is constructed using one standard deviation from the mean. *****; significance value of *p* = < 0.001.

**Figure 2 microorganisms-08-01026-f002:**
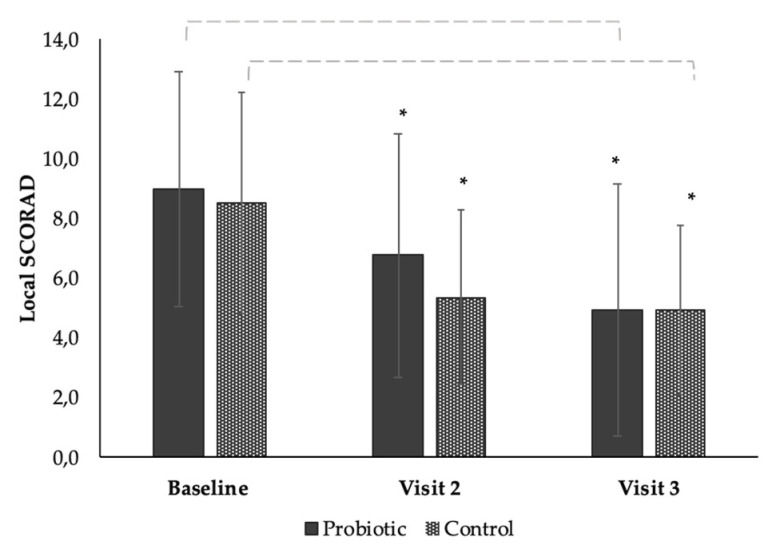
Bar chart showing the mean reduction in Local SCORAD of the subjects after continuous use of the probiotic or control product at baseline, visit 2 (day 28), and end of study visit (day 56). Each error bar is constructed using one standard deviation from the mean. *****; significance value of *p* = 0.001.

**Figure 3 microorganisms-08-01026-f003:**
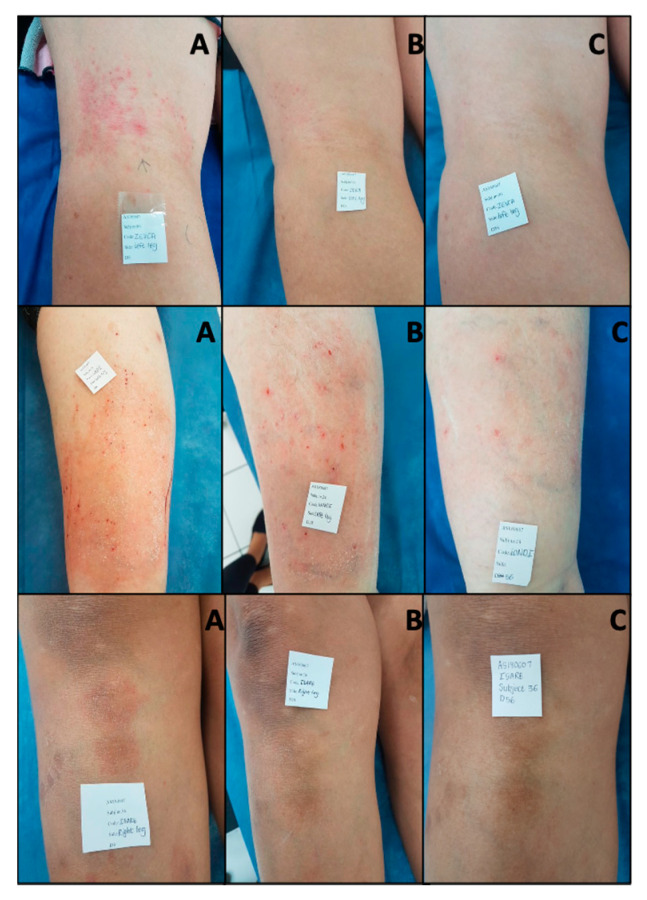
Pictures taken of present lesion of subject ^#^ 6 (Top row) ^#^ 24 (middle row) and ^#^ 36 (bottom row) in the probiotic product group from baseline (**A**), visit 2 (**B**), and end of study visit (**C**). Pictures were taken by the dermatologist investigator at the study clinic.

**Figure 4 microorganisms-08-01026-f004:**
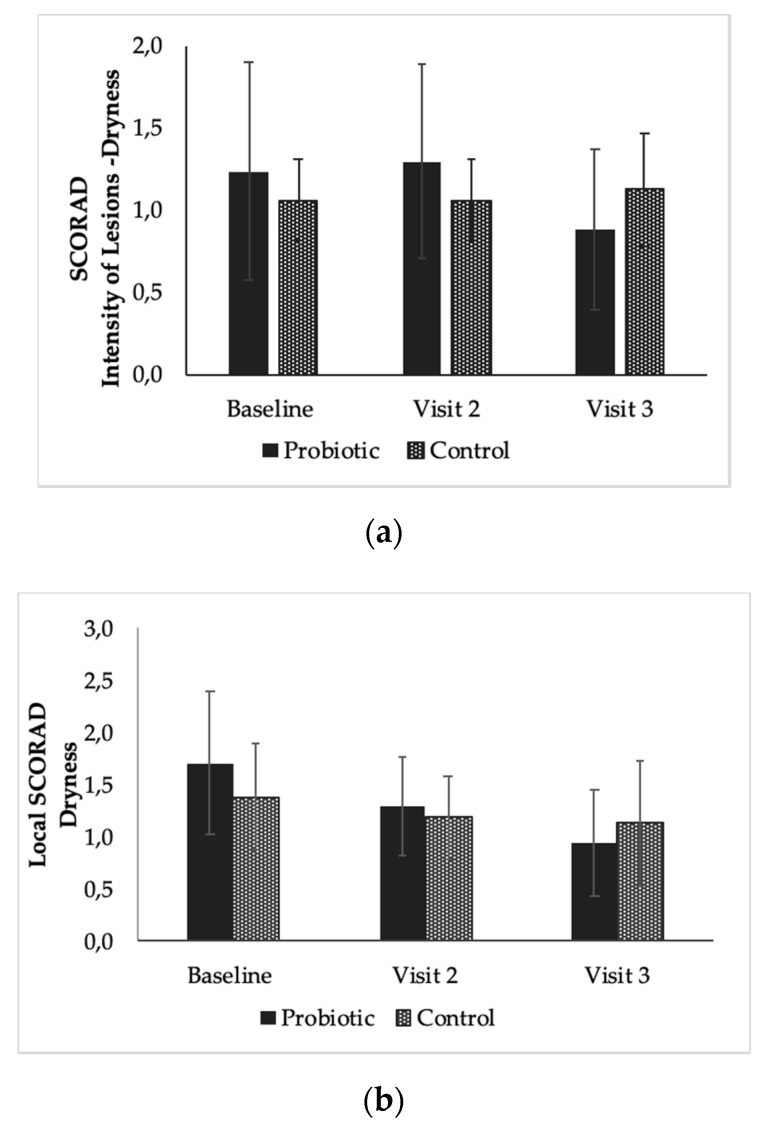
(**a**) Displays the mean dryness score of the intensity of lesions scores for dryness (SCORAD) of the subjects in each group for each time point. (**b**) Displays the mean dryness (local SCORAD) of the subjects for the probiotic product and the control product. Each error bar is constructed using one standard deviation from the mean.

**Table 1 microorganisms-08-01026-t001:** Table displaying the subject characteristics for each product group at inclusion; age, sex, ethnicity, skin appearance, Atopic Dermatitis (AD) severity, allergies, the average volume of used product. SCORing Atopic Dermatitis (SCORAD).

Parameter	Probiotic Group (*n* = 17)	Control Group (*n* = 17)
Age in years (range)	40.7 (19–66)	33.1 (19–57)
Female (*n*)	17	15
Ethnicity (*n*)		
*Caucasian*	7	6
*Black*	10	11
Skin Appearance		
*Normal*	2	2
*Dry*	12	9
*Very Dry*	3	6
AD severity at recruitment (SCORAD index) (x ± s.d)	41.7 ± 14.7	36.7 ± 10.7
Subjects with other allergies		
*Sinusitis*	2	2
*Hay fever, pollen, dust, pets*	3	6
Average volume of product used during study duration (g)	82.2 g	62.9 g

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
