# Peer review of "Lactobacillus reuteri DSM 17938 as a Novel Topical Cosmetic Ingredient: A Proof of Concept Clinical Study in Adults with Atopic Dermatitis"

_microorganisms, 2020, doi:10.3390/microorganisms8071026_

Round 1
Reviewer 1 Report
The Manuscript Number: microorganisms-842609 titled “Lactobacillus reuteri DSM 17938 as a novel topical cosmetic ingredient: a proof of concept clinical study in adults with Atopic Dermatitis” fit with the aim of the journal, it could suggest an important strategy for treatment of AD. Moreover, the authors present the product as full of advantages with respect other current used therapies. The paper can be published after the following major revisions. First of all the references should be update, including papers describing Lactobacillus reuteri DSM 17938 for its well characterized genotypic identity and its claims.
Moreover, it could be useful discuss about the skin microbiota features and the recent publication about the capability of strain belonging to the genus of Lactobacillus to inhibit, alone and/or in combination with other species commonly used as probiotic component in food supplements, the adhesion of both Gram positive and Gram negative pathogens, including Staphylococcus aureus. Other studies should be included considering the capability of Lactobacillus to produce antimicrobial metabolites able to inhibit the Gram positive and Gram negative pathogens growth.
Did the authors use the life cells of Lactobacillus reuteri DSM 17938 for the study? This point should be clarified in the text. It could be interesting including in vitro and ex vivo experiment, to justify the chose to use Lactobacillus reuteri DSM 17938 instead of a strain of Streptococcus salivarius or Lactococcus lactis, currently used as the probiotic strategy as probiotic strategy to rebalance skin microbiota.
It could be interested to know if the strain alive or its polysaccharide showed the same features in treatment of AD.
Lines 71-73: Please, motivate the choice of this specific paper and strain, considering the other paper present in literature and the different strain cited. Please, add other papers.
The authors are encouraged to revise the reference ad if it is possible the assays and re-submit soon the paper.
Author Response
Dear reviewer,
In response to your suggestions, we have included our answers to your comments below. We have included the relevant changes in the revised manuscript. Thank you once again for your consideration.
Best regards,
Jakob Axelsson
Strain characteristics: More details describing the characteristics of the chosen strain were added in terms of its genotypic identity and properties.
References: The references have been updated and missing references in the discussion have been added.
Skin microbiota: The inhibition of pathogens, including S. aureus, by L. reuteri and other Lactobacilli was elaborated on in the discussions section.
Live bacterial cells: Live probiotics were used in the investigational product. This has now been clarified in the abstract, introduction and section 2.2. The reason for using live probiotics was both discussed in the introduction by adding additional text.
Strain rational: The reason for choosing the probiotic strain in the active product is based on it’s characteristics in preclinical models. Additional text discussing this has been added in the introduction and its included reference.
Lines 71-73: The strain is not relevant for the paper and was removed and other references discussing the safety of the strain used in the paper, L. reuteri DSM 17938, was added.
Reviewer 2 Report
- I recommend that you shorten the introduction. This is a research article. Try to highlight only the important things in the introduction.
- "Emollient topic" is a general term. These include some dosage forms such as cream, ointments, lotion, etc. Which of these were used?
- Lines 84-86: “The aim of this study was to investigate and compare two investigational topical emollient products for cutaneous acceptability, safety and efficacy under normal conditions of use, in adult subjects with atopic dermatitis.”
Did you use two investigational products? Does that mean you don't have a control group? Why didn't you use a standard AD product (e.g., topical glucocorticoids)?
- Line 118: “Each subject had a two week wash out period where the subject could not modify their cosmetic/washing habits, use their standard topical treatment or apply cosmetics on the concerned areas.”
A washout period is defined as the time between treatment periods. So, it is important to know how many patients were under treatment when they were enrolled in the study. What type of treatment did they follow? Local or systemic? If they used systemic treatment what were the drugs? Was this 2-week period properly chosen?
- When did you calculate the SCORAD index for the inclusion of patients in the study? Before or after the wash out period?
- Line 124 -127: Regarding the exclusion criteria: did the participants have comorbidities? Did they receive any treatment for them? Could the treatment have influenced the results of this study? Were they excluded from the study?
- I think it is important to explain why you determined both the local SCORAD and the SCORAD index.
- I think you can compress the information in points 2.4 and 2.5 into a single paragraph.
- Line 152: “SCORing Atopic Dermatitis (SCORAD)”
I recommend that you explain the abbreviation when you first use the term in the text, not later.
- Line 156: Dermatologist – it is not necessary to use capital letter.
- Line 166: “A p-value below 5% was considered significant”
It is usually preferable to write 0.05 instead of 5%.
- Line 167-168: You are referring here to visit 2, visit 3, but so far you have not mentioned what they mean. Explain this in line 224. Why later?
- Lines 177-180: It is not necessary to repeat the main objectives in the results.
- Lines 182-187: Please be clear. I recommend that you rephrase this paragraph.
- Is it possible that the results include a table with the average value ± SD of SCORAD?
- Line 278: “Atopic Dermatitis (AD)” I recommend that you explain the abbreviation when you first use the term in the text, not later.
- Discussions: In my opinion, you insist here on the results obtained in other studies. You need to insist on your results and compare them with those obtained in other studies. I recommend you to review this section.
- The limits section should be clearer and it is not recommended to have references here.
- Lines 377-378: I recommend giving up this sentence. It's a general phrase. You must emphasize the conclusion of your study.
Author Response
Dear reviewer,
In response to your suggestions, we have included our answers to your comments below. We have included the relevant changes in the revised manuscript. Thank you once again for your consideration.
Best regards,
Jakob Axelsson
The introduction has been shortened according the reviewer’s suggested, as well as condensed and focusing more on fewer key aspects.
The product used is an ointment. This has been changed and specified throughout the text.
Line 84-86: The products used were one active product containing live probiotics and one control product containing other active ingredients but no probiotics. The wording “compare two investigational topical emollient products” has been changed to “compare a topical ointment product with and without live probiotic bacteria”.
In the clinical study, half of the participants were given the active product containing live probiotics and the other half belonging to the control group were given the control product. This has been updated and clarified throughout the manuscript.
The reason for investigating a probiotic and not using a standard AD product is due to side effects of the latter, and to observe the single effect of the investigational product, not its combined effect. A discussion on this has been added to the introduction.
Line 118: “Two week washout period” is an incorrect term to describe this period and should be “Two week run in period”. This has been changed throughout the manuscript.
Time of SCORAD index calculation: The SCORAD index for the inclusion of patients were calculated on baseline (day 0). It has been specified in section 2.3.
Line 124-127: The patients did not have any known co-morbidities that could influence results of this study such as immunological disease for example, nor did they receive any treatment for this. They were excluded from the study and the exclusion criteria (both diseases and medication) have been updated in the manuscript.
An additional sentence explaining the reason for including both local SCORAD and SCORAD index was added to section 2.3.
2.4 and 2.5: Section 2.4 and 2.5 were compressed into a single paragraph according to the reviewer’s suggestion.
Line 152: Text was added to explain SCORAD when first written in the manuscript (section 2.3) and was removed from line 152.
Line 156: Dermatologist, was changed to being spelled with small letters.
Line 166: “A p-value below 5%” was changed to “A p-value below 0,05” according to the reviewer’s suggestion.
Line 167-168: Definition of study visits was added in section 2.3.
Lines 177-180: The main objectives were removed from the results section.
Lines 182-187: The paragraph on protocol violations was clarified.
Tables A2 and A3: Tables showing the data (average values and SD) and p-values for SCORAD index and local SCORAD were added and references inserted into section 3.3 and 3.4.
Line 278: The abbreviation AD is being explained when first used in the introduction. The explanation of the acronym has been removed.
Discussion: Changes have been made to put more emphasis on the results of the current study and comparing them to previous finding and not the other way around.
Limits section: This section was clarified, and the literature reference was removed as suggested by the reviewer.
Lines 377-378: Sentence was removed according to reviewer’s suggestion.
Round 2
Reviewer 1 Report
Dear Authors,
thank you for the manuscript and its revisions. Now, it can be accepted as it.
Best regards
Author Response
Dear reviewer,
Thank you for your feedback.
Best regards/
Jakob Axelsson
Reviewer 2 Report
The article has been improved, but without impact on the abstract. Please improve this part of the article as well.
Use the same name in the article, either S. aureus or Staphylococcus aureus…..
Author Response
Dear reviewer,
Abstract: The abstract has been revised as suggested by the reviewer to reflect the changes that has been done to the manuscript as a whole. More precicely, it has been 1) shortened and made more to the point, 2) focused more on the used probiotic and 3) the two products were explained better.
Abbreviations: Abbreviations of bacterial latin names were made consistent throughout the manuscript. Staphylococcus, as well as Lactobacillus, was spelled out first time mentioned in the abstract as well as in the main text and was then consistently referred to as S. or L. respectively.
Best regards/
Jakob Axelsson